# A Sentiment Analysis Method Based on a Blockchain-Supported Long Short-Term Memory Deep Network

**DOI:** 10.3390/s22124419

**Published:** 2022-06-11

**Authors:** Arif Furkan Mendi

**Affiliations:** 1HAVELSAN, Information and Communication Technologies, 06510 Ankara, Turkey; afmendi@havelsan.com.tr or ariffurkan.mendi@ostimteknik.edu.tr; 2Department of Computer Engineering, Ostim Technical University, 06370 Ankara, Turkey

**Keywords:** sentiment analysis, blockchain, smart contracts, machine learning

## Abstract

Traditional sentiment analysis methods are based on text-, visual- or audio-processing using different machine learning and/or deep learning architecture, depending on the data type. This situation comes with technical processing diversity and cultural temperament effect on analysis of the results, which means the results can change according to the cultural diversities. This study integrates a blockchain layer with an LSTM architecture. This approach can be regarded as a machine learning application that enables the transfer of the metadata of the ledger to the learning database by establishing a cryptographic connection, which is created by adding the next sentiment with the same value to the ledger as a smart contract. Thus, a “Proof of Learning” consensus blockchain layer integrity framework, which constitutes the confirmation mechanism of the machine learning process and handles data management, is provided. The proposed method is applied to a Twitter dataset with the emotions of negative, neutral and positive. Previous sentiment analysis methods on the same data achieved accuracy rates of 14% in a specific culture and 63% in a the culture that has appealed to a wider audience in the past. This study puts forth a very promising improvement by increasing the accuracy to 92.85%.

## 1. Introduction

Mood is a feeling that dominates a person at a certain time and exhibits a long-term persistence. In addition to verbally expressing one’s emotions, mood analysis can also be carried out by observing behavior patterns. Mood is diverse by definition, as is emotion variety. In addition to the complexity and diversity of the emotional states of the person, there are also various states, such as depressive, euphoric (severe), angry and anxious [1].

Blockchain prioritizes data privacy through cryptographic privacy. With the transfer of any data following a transparent pathway, the starting block of the chain is formed. The approval mechanism works with the application of the transfer announcement covered by this block to every user in the network, which is considered as the “accounting ledger”.

The confirmation of this transfer by other users means that the “Smart Contract”, which is considered to be a block, applies the necessary action to be added to the chain. Various consensus mechanisms (Consensus) should be mentioned in this application.

Blockchain platforms, such as Bitcoin, which aims to move financial assets away from a central structure, and Ethereum [2], which allows data to differ from the central structure, provide this approval with the “Proof of Work” consensus mechanism [3].

However, today, many mechanisms have been formed. These reconciliation methods should be better than the efficiency values obtained by the central structures or should be able to achieve the same level of results. Making the best use of various costs and valuable factors such as energy, time, labor and natural resources are the requirements of reconciliation mechanisms.

Not only are they efficient, they also play a role in defining private, permissioned or public blockchain networks. Compromise methods are used in line with the purpose of the chain to be created. At this point, mechanisms such as “Proof of Stake”, “Proof of Delegated Stake”, “Proof of History”, “Proof of Authority” and “Proof of Identity”, which have been used more recently and popularly, can be mentioned [4]. Especially in areas where machine learning techniques are used, the proof of learning method emerges as an innovative method. This is a proposed mechanism for validating blocks of transactions in a distributed ledger inspired by machine learning competitions. These competitions help improve the state-of-the-art processes in many relevant tasks, such as image recognition, recommender systems, autonomous driving research, etc.

Blockchain, which operates using consensus methods, has the potential to work together in today’s application areas. The intellectual structure of the application provides this potential. The chain structure, which begins to form with the initial smart contract being posted to the ledger in block form, provides the right of immutability for each block included in the chain, if approved. With the inability to change each transparently presented block, the element of obscurity brought about by the central structure disappears [5].

Today, blockchain applications are carried out in the supply chain, especially in terms of food safety and precious metal tracking. It also contributes to the field of information technologies for political elections, market analysis and the development of business models. Blockchain can create a confirmation mechanism to reduce the margin of error in multi-layered research and analysis applications. Thus, the margin of error of simulations of various studies such as experimental and commercial applications can be reduced.

The confirmation of calculations can benefit valuable areas, such as workflow and forecasting. Initiatives for various sector benefits have been initiated with the aim of adopting this technology and nurturing the mentioned benefits. Thus, it has contributed to the establishment of transparent, traceable and efficient systems.

Due to the intensity of internet use and the speed of data flow, the fact that every development on a global scale is accessible to every individual connected to the internet, affects emotions and thoughts quickly and shapes behaviors. Considering the many psychological and physical interactions of daily lifestyle, it can be said that it has become quite complex.

In particular, the sensitivity factor, which is caused by the concentration of global data flow, is effective here. It is necessary to create more dynamic decision and action mechanisms in order to ensure adaptation to speed and situation by communicating any development instantly through communication channels. These influences are active in many areas, including private and professional life.

As has been mentioned, in human nature, since a reaction mechanism will come into play for every effect, emotional and intellectual changes are experienced by human beings. It is critical to evaluate the reflections of these changes as a result of sensors for commercial issues. Companies operating in data analysis make serious investments, especially in Big Data technologies. There are a variety of human behaviors included in large datasets. These behaviors can be analyzed by sending attitude data to the cyber environment through an electronic device participating in the internet environment.

Evaluating the behavioral attitudes of customers, especially in social and commercial operations, is a powerful study to reveal the value of data. Studies in this field are called sentiment analysis. This analysis method is an important part of approaches such as market research and customer service. They showcase ideas, feelings and tones on users’ overall experience. One of the methods currently used is the conversion of texts created by the combination of speaking style and tone of voice and their analysis [6]. However, the point to be noted here is that there are different types of reactions brought about by cultural differences.

When there is a standard evaluation of the data transferred in the communication channel of the cyber space and the human, serious problems can be encountered. While approximately 4 billion 660 million people worldwide can access the internet in 2021, and more than 4 billion 150 million people use social media [7]. Again, according to Statista’s data [8], it is seen that the most used language on the internet in 2020 is English, and approximately 26% of all users use the English language. Therefore, language and cultural differences should be considered in sentiment analysis. However, since English is used in the communication of commercial elements, which have gained popularity on a global scale, with customers, a sentiment analysis can be carried out within the largest market share, although it cannot appeal to all segments. The contents that are shared through the use of the mentioned communication channels and the effects of these contents on the buyer, can be used in this field as a result of the studies carried out with the blockchain activities, and the elements that can provide social and personal benefit are determined and suggestions are presented in line with the research. This study aimed to improve the results obtained as a result of traditional sentiment analysis by taking strength from the multi-layered blockchain structure of sentiment analysis. Thus, it aimed to achieve the following:Facilitating the analysis of changes in attitudes brought about by cultural differences, contributing to the correct evaluation of the behavioral and reaction-based effects of a mother tongue,Handling the data for mood management models, which are included in Big Data management models and whose value effect can be observed in many areas, correctly, thus personal relevance and user satisfaction are maximized,Meeting commercial and social expectations, increasing customer satisfaction and presenting the right product to the right user at the right time,

The purpose of mood research, which has the potential to protect users who are affected by the attraction of malicious activities in various sectors, is maintained with high success and high motivation.

As a new approach, the main contribution of this study to the literature of sentiment analysis is a supporting long short-term memory (LSTM) network with a Blockchain structure, including smart contracts. Our proposed approach achieves a very promising classification rate improvement.

## 2. Literature Review

Sentiment analysis refers to the application of natural language processing, computational linguistics and text analytics to identify and classify subjective opinions from among posts. In general, sentiment analysis aims to determine a content publisher’s attitude towards certain topics, such as the topic or contextual polarity of a post they share. Attitude can be defined as judgment or evaluation, emotional state or intended emotional communication. Sentiment analysis classifies textual expressions in source materials into two types. The first of these is facts (objective); includes objective statements about entities, events and their attributes (for example, “I bought an iPhone yesterday”). Another type is opinions (subjective). It covers subjective expressions of feelings, attitudes, evaluations or feelings towards entities, events and their qualities (for example, “I really like this new camera”).

It should be noted that not all subjective sentences contain ideas (For example, “I want a phone with good sound”), and not all objective sentences contain ideas (For example, “Headphone broke in just two days!”). Therefore, it is important to identify and extract facts and opinions from source text materials for sentiment analysis [9].

According to Zillman, mood management theory is the consumption of media messages and information according to the moods of individuals. It also assumes that an individual’s state of mind uses media influence and available options to optimize mood. He talks about trying to communicate with other people who have attitudes, skills, beliefs and knowledge. In case personal interests and necessary characteristics do not match, an “incompatibility” situation, which is a communication problem, is experienced. In order to avoid complete dissonance in the communication process, one will choose someone who exhibits opinions that are more relevant to one’s own area of interest. This is called “selective exposure”. This theory, which examines how individuals work to stay motivated in any negative mood, also examines how they manage to achieve the goal of high mood. The media tries to prepare the appropriate environment to meet the “hedonic” needs, which is the term meaning the pursuit of instant pleasure. The first random selection of media elements during the negative mood can activate the mood on the positive side. Exposure to entertainment elements that provide mood recovery movement after this breakdown makes it difficult to return to a negative mood. Thus, the mental interaction of the individual with a good mood is strengthened and an increase in daily activity performance occurs [10].

Media exposure is effective, although it is only one of many arousal options. While traditional tools of stimulus regulation often require considerable effort, such as a change of location or expending energy in an uplifting activity, they offer an opportunity for media exposure. Given that numerous media offerings are available almost everywhere, stimuli today are said to be more readily applicable than ever before. Initially, stimulus selection is random and independent of current mood states. However, choices made by chance during negative moods and successfully ending this harmful situation are negatively reinforced. Therefore, the probability and severity of impact will be higher in similar situations in the future. Similarly, exposure to stimuli that are successful in maintaining or intensifying good mood is positively reinforced, thereby increasing the efficacy in choosing similar stimuli in the future [11].

Sentiment analysis deals with analyzing the feelings and perspective of a speaker or author from a particular text. Sentiment analysis or opinion mining refers to the application of language, linguistics and textual analysis to detect and extract subjective information from procurement materials. This field of technology is concerned with analyzing and predicting background and obscure information. It gives valuable information about the users’ attitude, style and direction ratios on an example displayed in Figure 1.

Sentiment analysis is a specialized method of categorizing text in terms of subjective and objective nature. Judgment indicates that the text contains the content of ideas, while intelligibility indicates that the text does not contain the content of ideas. Accordingly, ideas are evaluated as positive or negative, objective or subjective [12].

Individuals often try to predict each other’s moods before interactions and to read each other’s moods during encounters. In these ways, mood information is acquired. This method is used informally to facilitate social and professional interactions. For example, knowing about the boss’ mood on a given day can help an employee anticipate the manager’s response to a call request. Similarly, information about consumers’ moods in marketing situations can provide marketers with a more complete learning about consumers and their responses to marketing strategies and tactics.

In terms of mood information, service encounters can be particularly critical to understanding consumer behavior as they are influenced by point-of-purchase stimuli, the content of marketing communications and the content covered by those communications. More generally, insights into consumer behavior can be gleaned by examining consumers’ thoughts and feelings. Significant research using traditional information processing values has enriched our understanding of the cognitive mediators of consumer behavior. Important information on consumer behavior has also been obtained from studies examining noncognitive (disbelief) factors such as subjective familiarity, entertainment and fantasy, motor and somatic representation of influence and emotion and attitude towards advertising. The findings show that these emotion-driven factors can play an important role in consumer attitude formation and brand selection.

Moods form a part of all marketing situations and can influence consumer behavior in many contexts, such as exposure to negative situations and brand choice. Mood can be quite fleeting and easily affected by little things. Small changes in the physical environment can affect consumers’ mood at the point of purchase. At the same time, small deviations in communication strategies can significantly affect their mood when exposed to advertising. In fact, while consumers’ moods are often influenced by factors beyond a marketer’s control, their mood can be greatly influenced by seemingly minor aspects of marketer behavior. Figure 2 includes the schematic representation of the emotional analysis parameters.

Mood can influence behavior during service encounters; good mood consumers may be more helpful and have a lower satisfaction threshold than neutral mood consumers. For example, moody consumers may be willing to delay gratification, follow doctor’s orders or pack their own food. On the other hand, service prescriptions cause good moods because they are less likely to act with expected negative outcomes, such as painful rehabilitation exercises or undergo medical tests that indicate illness. However, when the need for the behavior is clear, mood effects on performance can be minimal.

Material before an advertisement may include cognitive mood promoters, such as positive or negative statements, and noncognitive mood promoters, such as scary or happy music. While a full discussion of the techniques involved in the induction of emotions in media contexts is beyond the scope of this article, it should be noted that laboratory studies involving simple verbal messages have found that expressions in radio or newspaper reports can affect mood under conditions of forced exposure. The emotional content of the information presented in the studies was manipulated and hearing good/bad news was found to be associated with positive/negative mood measures. In addition, another study argued that self-reporting mood was found using newspaper reports about negative events. As illustrated in Figure 2, the effects of such inductive processes should be explored to understand the dissonance of moods relative to other mediators of context effects [13].

The analysis of human data is becoming commonplace in technology solutions. Using sentiment analysis, a subdiscipline of text analytics, offers organizations and individuals another dimension in understanding unstructured datasets such as Twitter and social feeds. When combined with real-time reporting, sentiment analysis can provide valuable contextual insight that enables the more precise interpretation of unstructured data. Incorporating this type of analysis into applications is a step towards cognitive IoT that can help improve decision making. Sentiment analysis is a subdiscipline of the broader field of text analytics. Exploring emotion within text or audio has always been an important part of text analysis, as it provides an extra dimension of understanding that goes beyond searching the text at the title level [14].

Blockchain and IoT-based performance measurement design in education makes it possible to evaluate the parameters of students from their mood to their success and transfer profiles. The main application of blockchain technology is the management of certificates. It includes processing and storing academic credentials such as transcripts, certificates, academic documents, degrees. The blockchain can be used to issue unique digital assets that validate the credentials of academic degrees and certificates. However, while data records will be visible to everyone, accessibility and modifications can only be made by authorized persons or organizations. In addition, IoT is committed to improving the quality of life of students and teachers. The information collected from the sensing devices helps to continuously monitor each student’s activities, mood, health, behavior and contribution to the class, thus allowing for fair assessment and generating student-centered response accordingly.

An efficient and fast teaching-learning process has been ensured and it has been determined with this study that the IoT factor plays a major role in this field. Thus, the advantage of the reliable and immutable side of the blockchain, which acts as a distributed digital ledger, comes to the fore. Thanks to applications in the education sector, it can be fully used to improve the entire teaching-learning process, ensure the fair evaluation of both students and teachers, improve performance and motivate students and teachers (giving prizes to students, management of records). These applications, which include innovative models and ideas, can pave the way for a better future for the education sector [15]. 

IBM Watson Tone Analyzer is an artificial intelligence program that uses linguistic analysis to detect emotional and language tones in written text. It analyzes the emotions and tones in what people write online, such as tweets or reviews, to predict whether they are happy, sad, confident and more. Furthermore, it monitors customer service and supports conversations to allow the appropriate and to scale response to customers. It analyzes whether the customers are satisfied and whether the agents are kind and sympathetic, allows the chat bot to detect customer tone so that the tone of the conversation can be set and dialogue strategies can be created. IBM Watson Natural Language Understanding uses deep learning to extract meaning and metadata from unstructured text data. It uses text analytics to detect categories, classification, entities, keywords, sentiment, sentiment, relationships and syntax [16].

## 3. Method and Applications

Sentiment analysis is a method used to gain insight into people’s social behaviors, ideas and feelings. It is also referred to as “Idea Mining”. As a result of mood detection algorithms and sensors, data are collected and a numerical classification of attitude process is performed. The study includes text-based analysis and the evaluation of Twitter-oriented personal content.

Sentiment analysis has a 3-layered working structure. This layer of work starts with the document level and is numbered positive, negative and neutral on the intelligibility of the attitude. Numbering operations are labelled as: –1 = Negative Emotional State, 0 = Neutral, 1 = Positive Emotional State.

In the next step, numerical classification is applied at the second layer, the sentence level analysis level, which is similar to document level. At the third level of appearance, various visual features, such as width and height, are classified. The visual layer application is not a layer application that can be used in every mood research.

It is not possible to examine the third layer, especially for personal content that includes text-based interaction and does not share a compatible communication channel.

The standard application image realized in each of these layers is shown in Figure 3.

Within the scope of emotion analysis, the “Emotion Detection” method adopts a study based on field dependent data. In line with the implementation of the “Emotion Detection” method, the “AIRC Sentiment Analyzer”, which operates on the basis of the English text language, and “RST Discourse Parsing”, one of the document-level emotion classification tools, were used. In addition, increasing the sensitivity of mood research is essential in mood research with low satisfaction levels. Therefore, adjustments have been made to classify syntaxes and identify high-impact infrastructure. It is aimed at increasing sensitivity by analyzing the syntaxes at the document level using the clustering method by dividing them into cores, thus making the sentiment analysis subject to strong results.

As stated in the introductory sequence, approximately 26% of all posts in the cyberspace can be addressed by conducting an English text-based mood research. There are text-based activities other than English, but they are more diverse and more costly. Since each work is addressed to a specific cultural audience, it is not realistic for the work to gain global popularity. One of the studies conducted in this regard was the trust criterion research, based on the Arabic culture, which exhibits different behaviors from the content that uses the English language culturally, and this difference can be felt in commercial applications, and where the behavioral habits differ and bilateral communication channels are evaluated. Social media communications need to be examined in order for the trust criterion to reach benchmarks. Nearly eleven million [18] of the restricted Arab cyberspace users communicate on text-based self-expression platforms. Interactions in these communication environments were examined with sequential and time-based trust statistics. Confidence statistics based on rank and time:Speech Confidence, the criterion calculated by considering the length, frequency and balance parameters of the communication between at least two users.Propagation Confidence, a measure of the frequency of message propagation between at least two users. The important indicator here is the concept of trust. It is assumed that the trust between the parties involved in the communication channel and the frequency of message spread are directly proportional. For example, if the first party trusts the second party, the first party will spread the messages from the second party and external interaction will increase. In addition, if the first party “Retweets” the second party’s post, it also means that the first party trusts the second party.

These criteria should be examined in a matrix based on the communication between the 2 parties. Accordingly, a trust value should be defined.

The generated square matrix is in the form of a simple symmetric matrix covering the communication channels created by four anonymous parties and is shown in Table 1.

Additionally, the sentiment analysis value (−1, 0 and 1) must be defined within the matrix.

The calculations must be recalculated for each topic in the communication channels and for each response given on each topic. Thus, as many matrices as the number of subjects will emerge. The S-value matrix calculation for Subject X is shown in Table 2.

Within the framework of the aforementioned studies, it is necessary to conduct text-based research by applying traditional mood research methods.

Two different infrastructure components were used in the simulation infrastructure of the designed system. The blockchain-based layer is built using the Hyperledger infrastructure. Hyperledger Fabric was used to establish the blockchain network, and Hyperledger Explorer, one of the tools provided by Hyperledger, was used to distribute and display blocks on the established network. All smart contract transactions are carried out over the Hyperledger Fabric layer. Matlab SIMULINK was used to perform sentiment analysis and data analytics with machine learning. Especially since machine learning processes require high performance, a need for a computer with high processing power has arisen. Especially for machine learning-based operations, a device with NVIDIA GeForce RTX 3080 graphics card, Intel i7 12700KF processor, 16GB RAM and 1TB SSD has been determined. In addition, Ubuntu 20.04 operating system is chosen to be used. 

### 3.1. Data Analysis

As mentioned earlier, Twitter is a very powerful text-based personal content sharing and communication channel creation platform. This platform is very popular and has more than 330 million active users [19]. Therefore, it is suitable for conducting sentiment analysis research. In this context, the social reactions of the effects that contain weights towards emotionality and expectations should be looked at. The data analysis started with the application of the trust criterion matrix, which was developed on the basis of cultural behavior reading created for a specific audience, which helps to show trust-based mood analysis in communication channels.

As a result of the examination of people who interact through Arabic language and culture within the scope of the study, mood analysis was performed based on 1, −1, 0 logical valuations, and the success rate was calculated as 11.6%, according to speaking confidence, and 14.4%, according to diffusion confidence. At this point, the biggest difficulties and deflectors experienced in the criterion parameters for sentiment analysis can be listed as the use of dialectical and contradictory Arabic, the lack of resources on attitude information and the separation of undesirable posts. For another study, which was comprehensively in English, the “2014 Apple Event” event, which was subjected to 1038 comments, was examined. The analysis of 28 random interpretations, which were samples, was carried out. The dataset for the study is given in Table 3.

The text-based data were subjected to English-based research by means of the mentioned document-based emotion management system “AIRC Sentiment Analyzer”. Parameters whose observance is considered critical for the application are: Number of emotion-based keywords,Number of followers,Number of follow-ups,Mood (0,1).

For the determination of keyword and mood numeric values, two attributes can be defined at the document level in the AIRC workbase:Expression of exaggeration,Informal writing expression,

The sentiment analysis results table, created by observing these qualities, is presented in Table 3 and Table 4.

The document-based operation implemented on the basis of the “AIRC Sentiment Analyzer” was 63% successful. Therefore, the transition from a specific culture to a behavioral culture that appeals to a wider audience has brought with it a more successful system with a wider audience and usable detection methods. It is necessary to repeat the number of replications in order to strengthen the AIRC system and give more accurate results. In the basic level trials with the “RST Discourse Decomposition” method for obtaining stronger replicants, the success rate increased to 65% for the existing dataset but decreased to 28% in the trials based on enhanced disaggregation. Table 5 and Table 6 include the application results of “RST Discourse Decomposition” and “Reinforced Discourse Decomposition”, respectively. As a result of this analysis, it can be seen that the success rates of mood studies that appeal to a mass culture are quite low compared to the masses with an intense language and culture, such as English. However, as a result of combined studies such as AIRC and RST, the results are not subject to correlation in English-based mood research. It is very critical to achieve a low success rate of 28% in reinforced applications. For this reason, it has emerged that blockchain applications should be based and applications that will play a role in a mood confirmation/confirmation mechanism should be distributed over multi-layered structures and create a decentralized mood structure.

### 3.2. Proposed Method

Sentiment analysis methods, in which the study structures and data evaluations mentioned in the text are made, are based on the practical applications of traditional research. It should be noted that mood research resorts to machine learning (ML) and applied machine learning to deep learning (DL) methods that work on classification sets. In order to analyze emotion, artificial neural network layers are used, and a category is formed by assigning a weight value to each layer. However, the results do not reach the desired success rate. For this, it is recommended to establish a multi-layered structure with a blockchain. In order to integrate the blockchain with mood research methods, long and short-term memory (LSTM) architecture, which is repetitive artificial neural networks established in deep learning, has been applied. LSTM differs from traditional deep learning with its feedback feature. The main purpose is to ensure that the blockchain, which strengthens the confirmation mechanism, can work backwards. Long and short-term memory can offer a study area for the past as well as the future in the structure of artificial neural networks. Figure 4 visualizes the artificial neural networks created using the classical deep learning method. Figure 5 shows the LSTM-based deep learning artificial neural network structure.

LSTM creates middleware layers to interpret one or more input values, then the numerical values made in these middlewares are sent to the output layer. Thus, the machine trying to learn an element begins to recognize it by assigning a numerical value to the element. This working structure is the basis of deep learning and explains why it is useful in the context of sentiment analysis. Although it is out of the question, LSTM supports the studies carried out on behalf of “Markov Chains”, one of the most valuable areas of probability and statistics applications. From this point of view, it can be concluded that deep learning applications are useful in string processing applications.

It is impossible for non-LSTM-based deep learning methods to perform a retrospective valuation check. Blockchain seeks a consensus structure between each middleware pair that enables fallback in LSTM-based network structures. For the creation of smart contracts, another layer in the artificial network structure is needed. Because, in order for evaluations to be processed in blocks, they must pass through a blockchain layer, where they must be processed into a function-based ledger. Blockchain decentralizes the storage of unmanaged or suspected data and contributes to machine learning by not making any retroactive changes. Transmitting the data that is sent reliably to the receiver, again, deals with the issue with a perspective that can minimize the error rate in the network structure. Because the data sent must be verified by the receiver, a confirmation layer is thus formed. Within the scope of the study, this layer is defined as the “Contract Layer”. A matching mechanism occurs when contract layers convert the data streamed in network structures in machine learning into a contract type, add the contract to the block, and keep the bit string value of the contract defined as the previous “block”. The machine algorithm, which improves the learning of the process over time, divides the harmony between the blocks into classes and follows its own data flow model. This monitoring process is able to act independently from the classification process made as a result of previous decisions and confirmations, because it is not possible for a smart contract to be corrupted, changed or removed from the chain structure.

In order for contract layers with a blockchain structure to work, the basic building blocks must be fully addressed. These can be specified as the blocks where smart contracts are defined, any data that passes through the network structure and converted into numerical valuation, the integrated node structure through which these data are streamed and the numbers that are the chain identity from which the classification score is created. In addition, it should not be forgotten that the basis of the blockchain is cryptographic encryption, which allows the binding of contracts. A working structure should also be specified for smart contracts to be completed and added to the chain structure. The point that needs to be highlighted in practice is to ensure that the network structure resulting from the integration of blockchain, and machine learning is more consistent than traditional blockchain structures. Because, as a result of the confirmation mechanism established with machine learning, the perception of trust that the ledger-structured database, in which the chain structure is processed, will increase. Thus, consistent blocks will be added to the ledgers created under a single classification name as a chain. The study was designed to have a structure shaped according to the first mood detection and the second mood detection. In this case, the primary decision maker for categorizing the blocks is the sentiment analysis result in the LSTM network structure. In other words, choosing which emotion class to be recorded in the chain ledger for the creation of the first block affects the entire process. It is necessary to gain the ability to evaluate the flow of the network structure not only forwards but backwards. The process after the initial block formation turns into a process that is fully under the control of the blockchain using the previous block classifications. Thus, inconsistent machine learning results are eliminated. In addition, a chain book is created for the subsequent evaluation of the data that cannot be included in the category, cannot be added to any previously defined block and cannot be matched to the chain to which the smart contract will be added, and a mood analysis program is created, which is free from duplications, missing values, data crowding and queuing problems. The principle that the blockchain structure will work in the system is shown in Figure 6. 

The model shows a flow categorized by machine learning as a result of the analysis of the data entered into the system. In order to keep the result in the database as a result of machine learning and to eliminate the mistakes that occur in the evaluation of new data entries to be exposed in the future, it is a concrete step to reducing the margin of error of the system by separating this learned category into ledgers and keeping the basic knowledge and knowledge (metadata) of these ledgers in the database. In order to fully define the identity of the ledger, a chain formation of at least two blocks should be followed, and simultaneous metadata transfer should be aimed for, through the formation of a block-binding cryptographic key. It should be emphasized that the number of ledgers is designed to be one more than the number of data types. The excess ledger contains unidentified data types. Additionally, data that cannot be emotion-identified is not added to the chains. The mechanism works in combination with deep learning. The main idea of the application is to eliminate the margin of error by creating an integrated chain structure that will support the accuracy of the analysis created as a result of special circumstances of the data shared by an individual with personal content and its compatibility with a personal content made by the individual.

The deep learning artificial neural network structure with the designed blockchain is shown in Figure 7. It defines text-based document-level content entered via composition, social media or message. Identified personal content is subjected to a numerical classification. The “Tokenization” process is applied for the classification activity. Inputs divided into cells are parsed in such a way that they are retained in the memory. Here, LSTM informal writing expression or exaggeration is sought. A dictionary value is assigned for each search item found. A text-digging activity is applied for every expression that is not included in the dictionary content registered in the databases. The text-digging activity works very hard, especially in the use of “Emoji” in messages created between communication channels [20]. In addition, machines that have performed these operations in the past and stored them in the database will directly apply word embedding if they have been exposed to the same message type. As a result of the mentioned stages, the data are brought together, and a numerical sentiment analysis is created, and attention is paid to the numerical expression of this sentiment. The numerical expression of emotion is defined as the “Class Score” (CS).

CS sent into the “First Contract Layer”, where the first step is taken for smart contract creation; it is evaluated in a decision mechanism. In order to create smart contracts at these layers, a blockchain system should be established. The blockchain system is important to define the internal structure of two intermediate and one final smart contract layers. The blockchain system is shown in Figure 8. The model includes a process that starts from the user layer, which is based on triggering the smart contract creation. The data entry, which is the smart contract trigger, is directly integrated into the personal content sharing platform on the machine learning network structure, and the smart contract is triggered with the login. The contract spark that initiates the formation of the blockchain structure transfers the transaction space to the proof layer. The evidence layer is given space for evidence that will enable contracting in the streams of data analysis that occur during machine learning. The confirmation mechanism to be created for the participation of the data passing through the artificial network structures to the smart contract is carried out with the necessary calculations to be mentioned, and the “Secure Matching” layer based on the consensus ideology is created. The mapping layer is an intermediate layer within the “Authority Layer” step. Here, the network structure in the account layer announces the success in return for the solution of the algorithm of decomposition of the SP value, which is the data classification process for the formation of smart contracts. Here, instead of a method, such as proof of work, that will cause a waste of energy, the proof of learning method, which has become popular thanks to its use in machine learning techniques and being a cost-effective verification method, has been used. Therefore, the contract is created using proof of learning, and the smart contract sent to the result layer is ready to be recorded in the ledger. The proof of learning method also helps to record the know-how of the contracts placed in the blockchain ledger in the machine learning database. As a result, consensus is achieved among the artificial networks, and the emotional classification is successfully recorded in the ledger.

CS can be −1, 0 or 1 as used in “AIRC Sentiment Analyzer” applications. However, in LSTM scaled blockchain applications, this value is provided in order to take any numerical value between −1 and 1. However, as a result of the calculation, the output number, which questions whether there is a mood, negative or positivity, is specified in the smart contract as 1, 0 or −1 output value without changing. The expression coefficient should be used for this calculation. The expression coefficient is obtained by the ratio of the classical basis logarithmic result of the cluster size and the sentiment weight value to the sum of the basis arithmetic results. Equation (1) covers the calculation of the expression coefficient and calculations of the base arithmetic value. In addition, Equation (2) shows how the CS value is calculated. The average indicator is an indicator value of loss and gain rates after “Tokenization”. CS values, which are polarized data, have to make use of the indicator.

CS = Class ScoreT = Class SizeW = Emotion Weight$lexS_{2}$ = Dictionary binary valueP = Expression CoefficientI = Average Indicator


(1)
$$Expression\;Coef\;\left(p\right)=\frac{\left(\log\left(T\times W\right)\right)}{\sum lexS_{2}}$$



(2)
$$CS=\frac{p}{\log\left(3\times W+I\right)}$$


The class score must have a validation value greater than the minimum threshold. This value is obtained from bitwise address holders that have been converted to digits by the use of computer language base arithmetic of dictionary entities pulled from the database. This highlights that each dictionary and text value corresponds to a number value in machine learning. Another parameter is the cluster size comparison of the numeric category of the emotion. It is the aim that this function is repeated more than once by taking advantage of the LSTM advantage and its smallest value is less than the SP value. Fulfilling this condition will fulfill the purpose of the first contract layer. If the condition result is confirmed as 0, the process is reset to search for a message entry other than the processed message, and the undefined blockchain is recorded in the ledger. Thmin (minimum transition threshold value) and ∆N (normalization change) are specified in Equations (3) and (4).
(3)$$∆N=\frac{p\times SP}{2-\log\left(T\times W\right)}$$
(4)$$Th_{min\;}=e^{\frac{∆N}{W+I}}$$

Upon the completion of the first phase of the smart contract, another message entry with the same sentiment category value of the evaluated user begins to be searched (Figure 9). A second sentimental message found is critical to establishing the notebook’s identity. After comparing the class score of the second message with the threshold value, it passes through the layers of the blockchain as a smart contract. In order to look at the compatibility of smart contracts, the compatibility of the SP value of the next input behavior with the SP value of the second behavior is searched for in the ledgers. The fact that the SP score of the second behavior is equal to the behavior that formed the previous contract indicates that that behavior also passed the threshold and was processed as a chain with cryptographic encryption in the same blockchain ledger. The equality can also be stated as a fit due to the values resulting from minimizing the error rates originating from LSTM. In this case, if compliance is achieved, an acquired compliance–confirmation mechanism is formed, thanks to the blockchain. The emotion validation mechanism allows the smart contract in the “Final Contract Layer” phase to be fully formed and posted to the ledger. If there is no harmony between any two SPs, a separate chain ledger is established for each emotion class data analyzed. Then the class data of the notebook is transferred to the database and used for subsequent mood determinations.

The chain structure, which is formed by repeating this confirmation system more than once, also helps to keep various statistics, such as the median, peak value and mode of mood research values. 

The application made to the network structure was evaluated in the dataset assessed within the scope of the study. As an example, it would be useful to process the process by considering a random content selected from the data set (Figure 10).

Since the blockchain technology is at the base of this study, the development phase of the system is started in line with the determined flow. In the proposed blockchain-based system, the proof of learning consensus method was adopted, the installation was carried out and the analysis was performed. An example screenshot where blocks, transactions and nodes are managed and viewed in the system that has been prepared is given in Figure 11.

In the proposed blockchain-based system, the SHA256 encryption algorithm was established and the transactions performed were encrypted thanks to this algorithm. The feature of SHA256 is that it converts data to a standard form and size. Data can be of different sizes and sizes. The result will always be the same size and structure. These data are 256 bits (32 bytes-64 hexadecimal) in size and is also called 256 hashes. In SHA256 encryption, data are converted to hash values, but this process is one-way, so hash values cannot be converted back to data. Another point to note here is that the hash values cannot be predicted due to their complex arrangement and that the results of very similar data are completely different from each other. Therefore, hash values are impossible to predict in advance. The slightest change in the data will cause the resulting hash value to be completely different. However, SHA256 is a deterministic operation. In other words, the calculation of hash values on the same data will always give the same result. Each transaction made in the established system is added to the distributed ledger as a new block. An example block representation in the installed system is given in Figure 12.

## 4. Results

Deep learning application with multi-layered blockchain application provides many contributions to the field where retrospective workspace can be presented with LSTM. The mood model that emerges as a result of feeding the sentiment analysis processes with the blockchain should optimally respond to today’s mood research purposes. The system and integration, designed to use the possibilities offered by blockchain on transparency and confirmation in deep learning steps, have exceeded the success levels specified in previous document-level sentiment analysis applications. The blockchain design implemented on the “Apple Event Data Set” table, displayed in Table 3, has been re-evaluated as the results shown in Table 7.

The sentiment analysis study with blockchain was concluded with a success rate of 92.857% on the final test.

This study is based on examining the effect of blockchain on mood research. Since traditional emotion research methods are based on discourse-based decompositions based on machine learning, there is a noticeable effect on the error rate. Because the artificial network structures of the deep learning method used by machine learning do not have a retrospective confirmation mechanism. For this purpose, using LSTM, “Short and Long-Term Memory”, discourse parsing, which is a dictionary value, recorded in the database is converted into symbols (Tokenization) and retrospective layer evaluation is performed. However, the LSTM method is insufficient in the use of lean. Creating a chain through smart contracts also provides the confirmation of retrospective LSTM assessments. Therefore, even if it is retrospective, it is designed to be an application that will greatly reduce the possibility of incompatible evaluations. However, the point that should be emphasized in this design is the need for knowledge and experience to result in numerical valuations of any message valued as input by a user for the mood management model, another mood behavior that should be included in the platform in the past or future time. In other words, the first mood analysis to be classified needs to define new contracts that will be added to the chain afterwards.

It is known that commercial activities for mood research increase efficiency, and in this context, it is used especially in the field of e-commerce. Determining the customer satisfaction of the operating parties and the effect of the product produced or sold on the final destination plays a very important role in drawing the way for their activities and updating the foresight. In this context, it was mentioned that the current study felt there was the need for additional data entry, due to the high success rate, additional time cost and the need for confirmation. In the study, which focuses on the fact that the success rates brought by the analysis of the content input alone are not satisfactory, a second same subject type new content input is required. One advantage of this is that it stretches the fields of study where traditional mood analyses have been constrained. It may go beyond standard applications such as text, audio, or visual- or color-based. Any deep learning algorithm can perform behavior classification with very high accuracy using the blockchain layer. The situation will support those parties with commercial and research activities in working more efficiently and more appropriately regarding the process. By minimizing the need for knowledge and experience for a solution to the negativities arising from the need for knowledge, establishing a contract structure in which unit content can be evaluated, and in order not to compromise the success rate of this contract structure, a calculation mechanism for the query threshold values of pseudo-copies of a single content can be created and subjected to the process. Thus, the first interaction process, which is the trial-and-error phase for the evaluation of the past behavior of a new party who has not shared a personal document level in the past, can be eliminated and the first impression of the party’s satisfaction can be contributed to the study by realizing the positivity. This copy process implies that various modifications must be made to the “Syntax” layer. At this point, a change in the working principle of the LSTM systematic dominates. It is also possible to use a different deep learning architecture, considering that the modifications to be made will cause other problems and damage the artificial network structure flow. Maintaining the coordinated flow of the deep learning artificial network layers is the desired modification result. In this way, in the process of activity based on a classical commercial basis, such as the marketing of a product, improvements can be achieved in various areas, such as ensuring the return of the product by the customer or a clearer and faster analysis of the comments made concerning the new products of the cinema industry, which is one of the leading fields of the entertainment industry. Meeting tempo and behavior coordination can be captured; customized advertising activities can be adopted for customers during the monitoring of commercial organizations, such as stores with real-time video analytics; and virtual assistant assets of products that are subject to artificial intelligence developments that lead to digital transformation, such as autonomous vehicles, can be customized according to the customer. There is no cultural influence dependency on the basis of the realization of these added values. In this respect, this study should be considered as a strong alternative.

The contributions of deep learning artificial network structure arrangement, which is a design in which the return of blockchain is felt, will benefit the development of new perspectives on data evaluations that can increase satisfaction in many areas, such as commercial, social, political and derivatives. The emergence of a correlation between the stakeholders will also be considered as a trigger for the increase in the economic efficiency of the sectors operating in data analysis.

Table 8 presents the data including the comparison of the results of AIRC and RST with the blockchain-based LSTM architecture, which is the proposed system design, in the same order as the other tables of the comments examined during the study. The “Sentiment State” column shows the mood determinations based on the personal inferences of the comments reviewed before the study. Mood determinations that do not comply with these values are highlighted with the color red, and success rates are handled in a concrete way. Thus, it can be presented as an ideal data visualization tool to see which mood detection method gives results in which situations. The results of sentiment state and blockchain-based LSTM result comparison are shown in Figure 13. It is clear that a real improvement is achieved with the blockchain-based LSTM case.

As can be seen in Table 4, Table 5 and Table 7, the keyword changes handled by mood detection methods affect the results obtained. As stated in Table 9, the differentiation of the processing of the contents by reducing them to keywords and their evaluation in network structures allows the evaluation of the working principles of the methods. Based on the mood results and the related success rates, the lack of a negative mood definition of the AIRC Sentiment Analyzer and RST discourse parsing methods, the improvements made in syntax parsing and the low success rate despite the increase in the number of replications indicate a network processing inconsistency. The metadata provided by the blockchain ledgers to the database, supported the analysis and evaluation of keywords in a flexible and clear way.

Blockchain-based LSTM architecture can solve the negative mood problem that AIRC and RST methods cannot achieve. In addition to producing flexible results, it produced results in line with expectations in 26 of 28 document-based contents, reaching a very high success rate of 92.85%. This important achievement could be seen in Figure 14, in which the comparison of sentiment state and all the other analyses is shown. According to the results indicated in red in Table 8, this comprises 9 erroneous results in the AIRC method and 11 erroneous results in the RST method. It can be seen that the erroneous results of the proposed design are due to the fact that the document samples containing mood are treated as documents that do not contain mood. At this point, it can be solved by the increase in the number of mood-smart contracts committed to the blockchain and the evaluation of similar keywords from the content at the syntax layer. As expected, the draft may have a margin of error according to the mood frequency, due to the fact that it has a systematic based on the confirmation of the individual with the individual. This data briefly presents the ratio of the replication enhancement and the resolution performance of the chain-based LSTM architecture.

## 5. Discussion

In this study a blockchain-supported LSTM architecture is proposed for sentiment analysis. Especially as the proposed approach is superior to the AIRC and RST methods in terms of predicting negative mood. In addition to producing flexible results, it achieves a very high classification rate of 92.85% for positive, neutral and negative moods. This shows that proposed approach can be successfully used for sentiment analysis. The most important contribution of this study to the literature is fusing a blockchain layer with LSTM layers. In the case of more complex data, in the future, the classification accuracy can be kept and improved by increasing the number of mood-smart contracts associated with the blockchain structure.

## Figures and Tables

**Figure 1 sensors-22-04419-f001:**
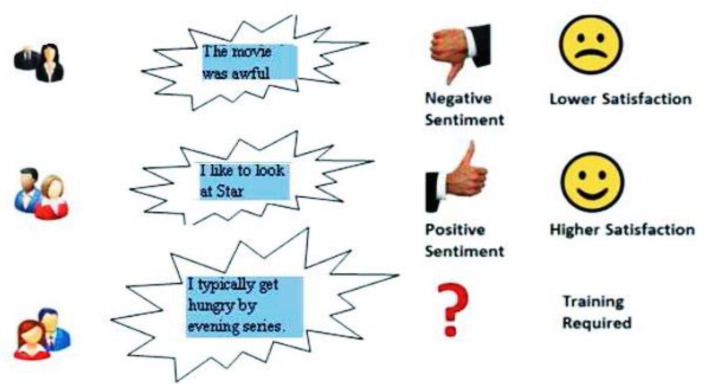
An example of emotional analysis results [12].

**Figure 2 sensors-22-04419-f002:**
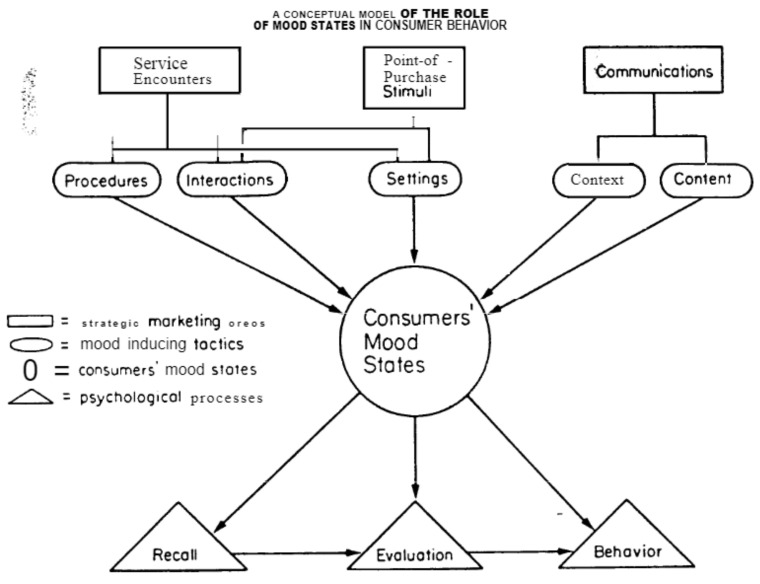
Content and Induction of Consumer Mood States [13].

**Figure 3 sensors-22-04419-f003:**
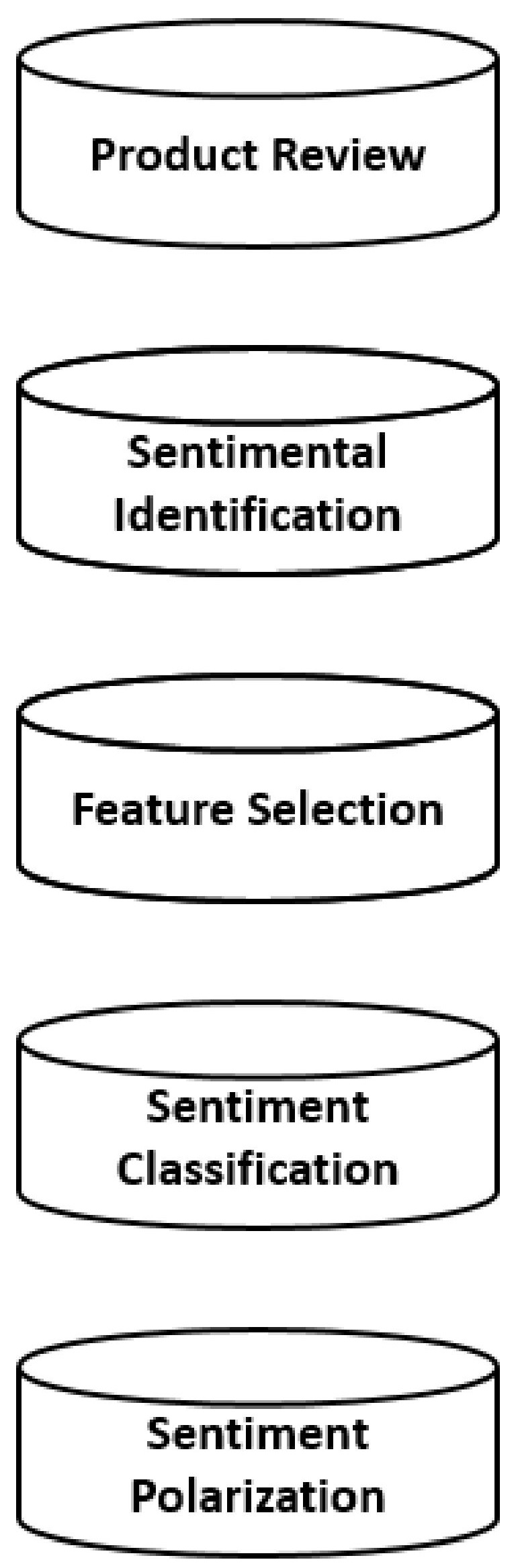
Layer based Study [17].

**Figure 4 sensors-22-04419-f004:**
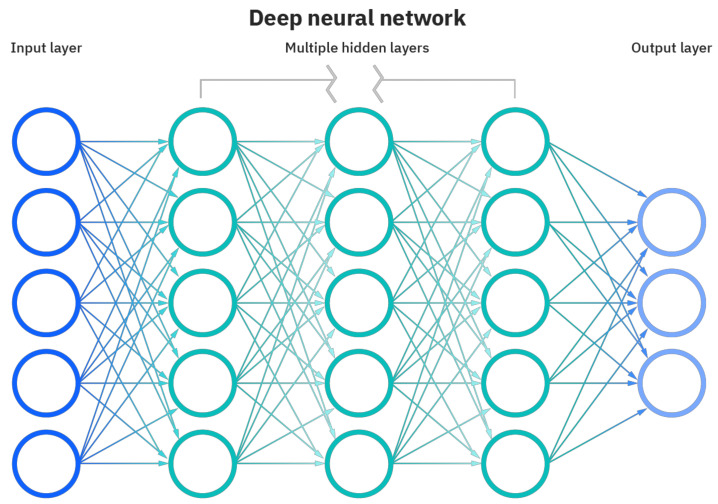
Deep neural network.

**Figure 5 sensors-22-04419-f005:**
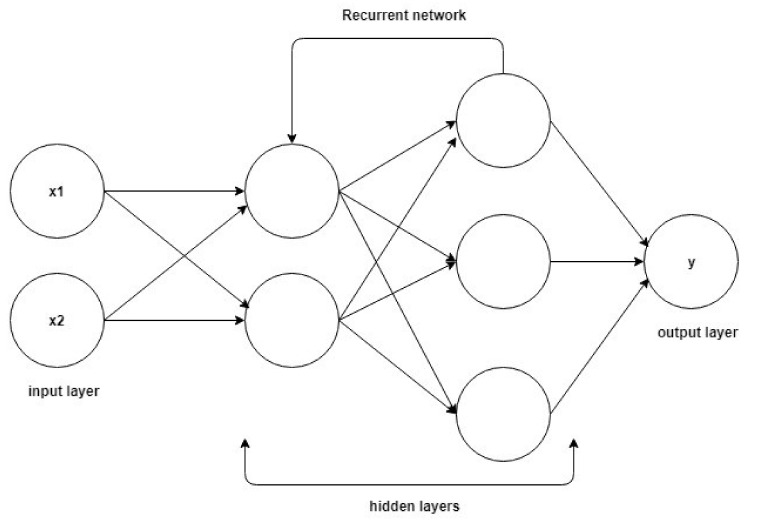
LSTM-based deep neural network.

**Figure 6 sensors-22-04419-f006:**
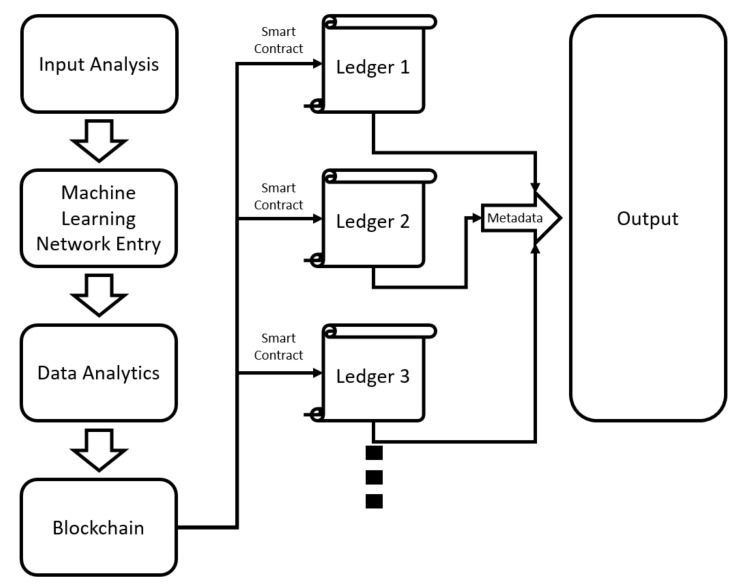
Machine learning and blockchain integration model.

**Figure 7 sensors-22-04419-f007:**
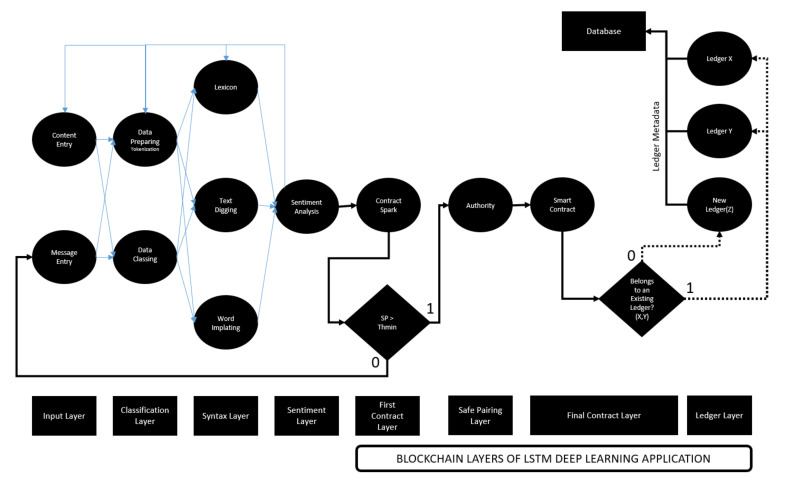
Proposed blockchain–LSTM neural network structure.

**Figure 8 sensors-22-04419-f008:**
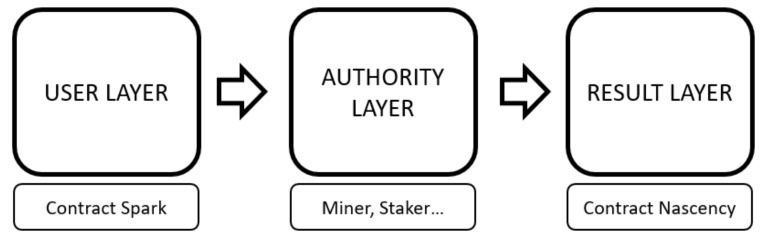
Blockchain layer-based process model.

**Figure 9 sensors-22-04419-f009:**
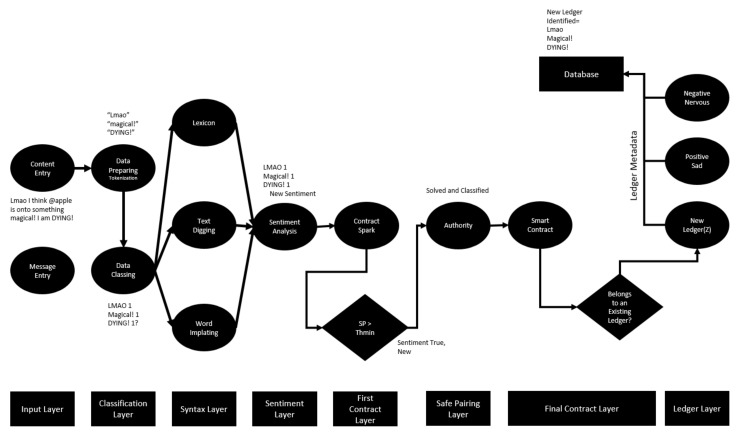
Newly defined sentiment entry and notebook formation data flow chart.

**Figure 10 sensors-22-04419-f010:**
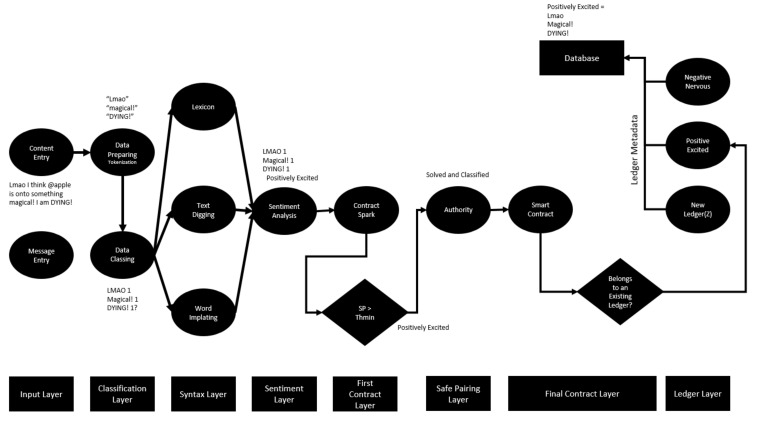
Random time, second least positive excited data flow chart.

**Figure 11 sensors-22-04419-f011:**
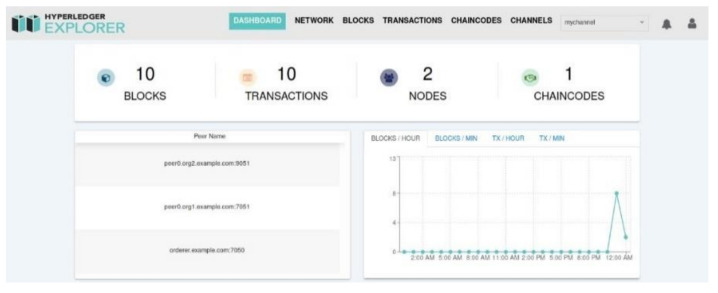
Monitoring screen of the established blockchain-based system.

**Figure 12 sensors-22-04419-f012:**
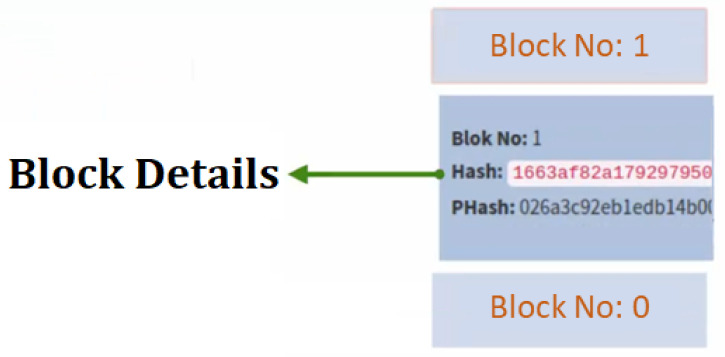
An example of block structure in the proposed blockchain-based system.

**Figure 13 sensors-22-04419-f013:**
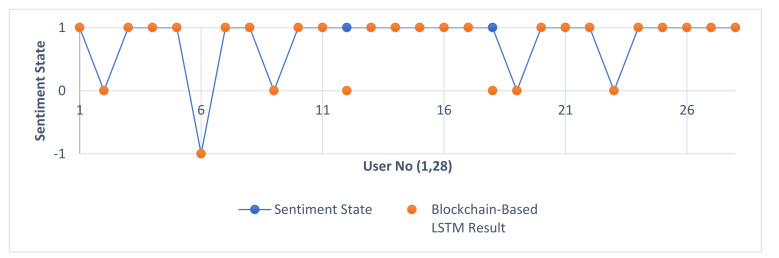
Sentiment state and blockchain-based LSTM comparison.

**Figure 14 sensors-22-04419-f014:**
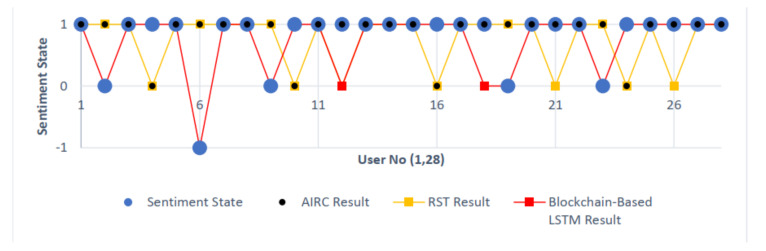
Sentiment state and all analysis comparison.

**Table 1 sensors-22-04419-t001:** Confidence Criterion Matrix [18].

	A	B	C	D
A	-	t_ab_	t_ac_	t_ad_
B	t_ba_	-	t_bc_	t_bd_
C	t_ca_	t_cb_	-	t_cd_
D	t_da_	t_db_	t_dc_	-

**Table 2 sensors-22-04419-t002:** Confidence Criterion Matrix for Subject X [18].

Topic x {x:1…n}	A	B	C	D
A	-	St_x_ab	St_x_ac	St_x_ad
B	St_x_ab	-	St_x_bc	St_x_db
C	St_x_ac	St_x_cb	-	St_x_cd
D	St_x_ad	St_x_db	St_x_cd	-

**Table 3 sensors-22-04419-t003:** Apple Event, Twitter Comment Data Set [17].

Username	Documents
Joel Comm	Now all @Apple has to do is get swype on the iphone and it will be crack Iphone that is
Vincent Boucher	@Apple will be adding more carrier support to the iPhone 4S (just announced)
William Tincup	Hilarious @youtube video-guy does a duet with @apple s Siri Pretty much sums up the love affair
Aaron Carter	@RIM you made it too easy for me to switch to @Apple iPhone. See ya!
BILLIONAIRE PR GIRL	I just realized that the reason I got into twitter was ios5 thanks @apple
Kim Garst	Im a current @Blackberry user little bit disappointed with it! Should I move to @Android or @Apple @iphone
#DeepLearning #App	The 16 strangest things Siri has said so far. I am SOOO glad that @Apple gave Siri a sense of humor!
Andrea Feczko	Great up close & personal event @Apple tonight in Regent St store!
Stephen Stephan	From which companies do you experience the best customer service aside from @zappos and @apple?
Neechi	Just apply for a job at @Apple hope they call me lol
#Talk2Me	RT @JamaicanIdler: Lmao I think @apple is onto something magical! I am DYING!!! haha. Siri suggested where to find whores and where to h
IG@MDoTMancini	Lmao I think @apple is onto something magical! I am DYING!!! haha. Siri suggested where to find whores and where to hide a body lolol
Kirby Ellis	RT @PhillipRowntree: Just registered as an @apple developer... Heres hoping I can actually do it... Any help greatly appreciated!
Lori Ruff	Wow. Great deals on refurbed #iPad (first gen) models. RT: Apple offers great deals on refurbished 1st-gen iPads @Apple
Ron Edmondson	Just registered as an @apple developer... Heres hoping I can actually do it... Any help greatly appreciated!
Marcus2braids	Just registered as an @apple developer... Heres hoping I can actually do it... Any help greatly appreciated!
AboveAverageClot hing	! Currently learning Mandarin for my upcoming trip to Hong Kong. I gotta hand it to @Apple iPhones & their uber useful flashcard apps
im n DALLAS	Come to the dark side @gretcheneclark: Hey @apple if you send me a free iPhone I will publicly and ceremoniously burn my #BlackBerry.
Samantha	Hey @apple if you send me a free iPhone (any version will do) I will publicly and ceremoniously burn my #BlackBerry.
The Product Poet	Thank you @apple for Find My Mac-just located and wiped my stolen Air. #smallvictory #thievingbastards
STEPH (OnHol)// ifb	Thanks to @Apple Covent Garden #GeniusBar for replacing my MacBook keyboard/cracked wristpad during my lunch break today out of warranty.
Calvin Lee	@DailyDealChat @apple Thanks!!
Elnor Bracho	iPads Replace Bound Playbooks on Some N.F.L. Teams@apple @nytimes
J’Corey Lamar	@apple..good ipad
Maliachi Broadwater	@apple @siri is efffing amazing!!
IG: @JabariStaffz	Amazing new @Apple iOs 5 feature.jatFVfpM
Gordon Tredgold	RT @TripLingo: Were one of a few Featured Education Apps on the @Apple **Website** today sweet!
Bill Hibbler	Were one of a few Featured Education Apps on the @Apple **Website** today sweet!

**Table 4 sensors-22-04419-t004:** AIRC Sentiment Analyzer Result [17].

Username	Keyword Repetitions	Follower/Following Ratio	Sentiment
Joel Comm	1	1.00	1
Vincent Boucher	3	1.39	1
William Tincup	3	0.92	1
Aaron Carter	1	1.71	0
BILLIONAIRE PR GIRL	1	1.41	1
Kim Garst	1	1.63	1
#DeepLearning #App	1	1.81	1
Andrea Feczko	1	1.07	1
Stephen Stephan	4	1.00	1
Neechi	1	2.39	0
#Talk2Me	4	2.72	1
IG@MDoTMancini	3	0.97	1
Kirby Ellis	1	1.31	1
Lori Ruff	1	1.00	1
Ron Edmondson	1	1.16	1
Marcus2braids	1	1.00	0
AboveAverageClot hing	1	0.94	1
im n DALLAS	2	1.01	1
Samantha	3	1.17	1
The Product Poet	1	1.51	1
STEPH (OnHol)// ifb	3	1.32	1
Calvin Lee	1	0.97	1
Elnor Bracho	1	2.12	1
J’Corey Lamar	1	1.25	0
Maliachi Broadwater	10	1.00	1
IG: @JabariStaffz	3	1.60	1
Gordon Tredgold	1	2.00	1
Bill Hibbler	1	1.18	1

**Table 5 sensors-22-04419-t005:** Mood Research Results with RST decomposition method [17].

Username	Keyword Repetitions	Follower/Following Ratio	Sentiment
Joel Comm	1	1.00	1
Vincent Boucher	3	1.39	1
William Tincup	3	0.92	1
Aaron Carter	1	1.71	0
BILLIONAIRE PR GIRL	1	1.41	1
Kim Garst	1	1.63	1
#DeepLearning #App	1	1.81	1
Andrea Feczko	1	1.07	1
Stephen Stephan	4	1.00	1
Neechi	1	2.39	0
#Talk2Me	4	2.72	1
IG@MDoTMancini	3	0.97	0
Kirby Ellis	1	1.31	1
Lori Ruff	1	1.00	1
Ron Edmondson	1	1.16	1
Marcus2braids	1	1.00	0
AboveAverageClot hing	1	0.94	1
im n DALLAS	2	1.01	1
Samantha	3	1.17	1
The Product Poet	1	1.51	1
STEPH (OnHol)// ifb	3	1.32	0
Calvin Lee	1	0.97	1
Elnor Bracho	1	2.12	1
J’Corey Lamar	1	1.25	0
Maliachi Broadwater	10	1.00	1
IG: @JabariStaffz	3	1.60	0
Gordon Tredgold	1	2.00	1
Bill Hibbler	1	1.18	1

**Table 6 sensors-22-04419-t006:** Mood research results using the enhanced RST discrimination method [17].

Username	Keyword Repetitions	Follower/Following Ratio	Sentiment
Joel Comm	3	1.00	1
Vincent Boucher	4	1.39	1
William Tincup	0	0.92	0
Aaron Carter	2	1.71	1
BILLIONAIRE PR GIRL	0	1.41	0
Kim Garst	0	1.63	0
#DeepLearning #App	0	1.81	0
Andrea Feczko	0	1.07	0
Stephen Stephan	0	1.00	0
Neechi	0	2.39	0
#Talk2Me	3	2.72	1
IG@MDoTMancini	3	0.97	1
Kirby Ellis	2	1.31	0
Lori Ruff	2	1.00	1
Ron Edmondson	0	1.16	0
Marcus2braids	1	1.00	1
AboveAverageClot hing	2	0.94	1
im n DALLAS	3	1.01	1
Samantha	0	1.17	0
The Product Poet	0	1.51	0
STEPH (OnHol)// ifb	0	1.32	0
Calvin Lee	1	0.97	1
Elnor Bracho	1	2.12	1
J’Corey Lamar	0	1.25	0
Maliachi Broadwater	0	1.00	0
IG: @JabariStaffz	0	1.60	0
Gordon Tredgold	3	2.00	1
Bill Hibbler	1	1.18	0

**Table 7 sensors-22-04419-t007:** Result of proposed method on Apple event dataset.

Username	Keyword Repetitions	Follower/Following Ratio	Sentiment
Joel Comm	1	1.00	1
Vincent Boucher	0	1.39	0
William Tincup	2	0.92	1
Aaron Carter	1	1.71	0
BILLIONAIRE PR GIRL	1	1.41	1
Kim Garst	3	1.63	−1
#DeepLearning #App	2	1.81	1
Andrea Feczko	1	1.07	1
Stephen Stephan	4	1.00	1
Neechi	1	2.39	0
#Talk2Me	6	2.72	1
IG@MDoTMancini	6	0.97	1
Kirby Ellis	1	1.31	0
Lori Ruff	1	1.00	1
Ron Edmondson	1	1.16	1
Marcus2braids	1	1.00	0
AboveAverageClot hing	1	0.94	1
im n DALLAS	3	1.01	0
Samantha	3	1.17	1
The Product Poet	1	1.51	1
STEPH (OnHol)// ifb	2	1.32	0
Calvin Lee	1	0.97	0
Elnor Bracho	1	2.12	−1
J’Corey Lamar	0	1.25	0
Maliachi Broadwater	1	1.00	1
IG: @JabariStaffz	1	1.60	1
Gordon Tredgold	1	2.00	1
Bill Hibbler	1	1.18	1

**Table 8 sensors-22-04419-t008:** Benchmark of data set evaluation results.

Username	Sentiment State	AIRS Result	RST Result	Blockchain-based LSTM Result
Joel Comm	1	1	1	1
Vincent Boucher	0	1	1	0
William Tincup	1	1	1	1
Aaron Carter	1	0	0	1
BILLIONAIRE PR GIRL	1	1	1	1
Kim Garst	−1	1	1	−1
#DeepLearning #App	1	1	1	1
Andrea Feczko	1	1	1	1
Stephen Stephan	0	1	1	0
Neechi	1	0	0	1
#Talk2Me	1	1	1	1
IG@MDoTMancini	1	1	0	0
Kirby Ellis	1	1	1	1
Lori Ruff	1	1	1	1
Ron Edmondson	1	1	1	1
Marcus2braids	1	0	0	1
AboveAverageClot hing	1	1	1	1
im n DALLAS	1	1	1	0
Samantha	0	1	1	0
The Product Poet	1	1	1	1
STEPH (OnHol)// ifb	1	1	0	1
Calvin Lee	1	1	1	1
Elnor Bracho	0	1	1	0
J’Corey Lamar	1	0	0	1
Maliachi Broadwater	1	1	1	1
IG: @JabariStaffz	1	1	0	1
Gordon Tredgold	1	1	1	1
Bill Hibbler	1	1	1	1
Classification Accuracy Avg.	-	67.85%	57.14%	92.85%

**Table 9 sensors-22-04419-t009:** Benchmark of sentiment data set keywords evaluation results.

Username	Sentiment State	AIRS Keywords	RST Keywords	Blockchain-based LSTM Keywords
Joel Comm	1	1	1	1
Vincent Boucher	0	3	3	0
William Tincup	1	3	3	2
Aaron Carter	1	1	1	1
BILLIONAIRE PR GIRL	1	1	1	1
Kim Garst	−1	1	1	3
#DeepLearning #App	1	1	1	2
Andrea Feczko	1	1	1	1
Stephen Stephan	0	4	4	4
Neechi	1	1	1	1
#Talk2Me	1	4	3	6
IG@MDoTMancini	1	3	1	6
Kirby Ellis	1	1	1	1
Lori Ruff	1	1	1	1
Ron Edmondson	1	1	1	1
Marcus2braids	1	1	1	1
AboveAverageClot hing	1	1	1	1
im n DALLAS	1	2	2	3
Samantha	0	3	3	3
The Product Poet	1	1	1	1
STEPH (OnHol)// ifb	1	3	3	2
Calvin Lee	1	1	1	1
Elnor Bracho	0	1	1	1
J’Corey Lamar	1	1	1	0
Maliachi Broadwater	1	10	10	1
IG: @JabariStaffz	1	3	3	1
Gordon Tredgold	1	1	1	1
Bill Hibbler	1	1	1	1

## Data Availability

The data presented in this study are available on request from the corresponding author.

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
