# Peer review of "A Sentiment Analysis Method Based on a Blockchain-Supported Long Short-Term Memory Deep Network"

_sensors, 2022, doi:10.3390/s22124419_

Round 1
Reviewer 1 Report
- This paper does not see the specific application of blockchain technology, and the description of this part can be added.
- How do you apply the hash algorithm SHA256? This paper has not seen it. The author's technical research is not deep enough. This can be explained again.
- Please explain the relationship between Proof of Learning and blockchain.
- Whether there is a system to make a screen, you can take a screenshot to explain.
Reviewer 2 Report
The paper reflects substantial work. However, it has significant flaws in the proposed method. Please refer to my comments on the document.
- The author should improve his PoC by using graphs
- He should describe his simulation environment
The abstract should be revised to reflect the paper’s contents.
Round 2
Reviewer 1 Report
This author has answered previously posed questions. The paper becomes more complete. The author has revised the paper well.
Reviewer 2 Report
Thank you for addressing most of my concerns.